# Work-Family Conflict in the European Union: The Impact of Organizational and Public Facilities

**DOI:** 10.3390/ijerph16224419

**Published:** 2019-11-12

**Authors:** Chantal Remery, Joop Schippers

**Affiliations:** 1Utrecht University School of Economics, Utrecht University, P.O. Box 80125, 3508 TC Utrecht, The Netherlands; 2Faculty of Law, Economics and Governance, Utrecht University, P.O. Box 80125, 3508 TC Utrecht, The Netherlands; j.j.schippers@uu.nl

**Keywords:** work-family conflict, work-family facilities, comparative study, European Union, gender

## Abstract

Today, as an increasing share of women and men is involved in both paid tasks at work and unpaid care tasks for children and other relatives, more people are at risk of work-family conflict, which can be a major threat to well-being and mental, but also physical health. Both organizations and governments invest in arrangements that are meant to support individuals in finding a balance between work and family life. The twofold goal of our article was to establish the level of work-family conflict in the member states of the European Union by gender and to analyze to what extent different arrangements at the organizational level as well the public level help to reduce this. Using the European Working Conditions Survey supplemented with macro-data on work-family facilities and the economic and emancipation climate in a country, we performed multilevel analyses. Our findings show that the intensity of work-family conflict does not vary widely in EU28. In most countries, men experience less work-family conflict than women, although the difference is small. Caring for children and providing informal care increases perceived work-life conflict. The relatively small country differences in work-family conflict show that different combinations of national facilities and organizational arrangements together can have the same impact on individuals; apparently, there are several ways to realize the same goal of work-family conflict reduction.

## 1. Introduction

Until the 1970s, most Western countries were characterized by a simple division of labour. Men acted as breadwinners for the whole family and did most of the paid work in the labour market, while women acted predominantly as mothers and housewives, taking care of the children, all kinds of household chores and, if necessary, also looking after members of the extended family in need of care. If women were active in the labour market, it was often before they got married and had family responsibilities, and in typical female jobs like that of a secretary, behind the counter as a shop assistant or in a factory performing different sorts of repetitive tasks. The last quarter of the 20th century welcomed the emancipation of women and the emergence of married women in the labour market. Some Western countries were frontrunners (e.g., the Scandinavian countries), while others lagged behind (e.g., some of the family-oriented Mediterranean countries). However, the trend was clear; more and more women became active in two domains of life: their traditional domain of care and the family, and their newly conquered domain of paid work in the labour market [1,2]. Even though this development can be characterized as a major success of emancipation and many economies took great benefits from additional labour supply and the contribution of female talent [3], the new role of women also gave rise to a new phenomenon: work-family conflict [4]. Combining tasks in both domains of life does not always run smoothly. 

Work-family conflict (WFC) is primarily a matter of competing time claims [5,6]. First, the total work and family tasks may exceed the available time budget of twenty-four hours in a day (of which one has to deduct, of course, several hours for sleep and personal care). Secondly, WFC may follow from intertemporal problems: one must attend a meeting or give a lecture at the same time as one would like to accompany a parent to the hospital, pick up the children from school or be at home when the plumber comes to repair the leaking roof. Work and family tasks require people to be in different places at the same time. Previous research has shown that women, next to their paid work, still perform the majority of household and care tasks for children and other relatives [7] and even more often, take the responsibility for the organisation of those tasks [8,9,10].

Yet, WFC has not left men unaffected. The time men spend on household and caring tasks has increased [11,12]. On the one hand, modern emancipated women expect and even demand the fathers of their children to contribute actively to the upbringing of their children, while on the other hand, modern emancipated men realize that there is more to life than a successful job career [13]. Thus, increasingly—and this especially holds for the younger generations—men also combine paid work and care tasks for their children [14]. Consequently, they also fall prey to the risk of experiencing WFC [15].

For both women and men, this risk may aggravate due to the growing need for informal care from the older generations. All over Europe, the share of elderly is increasing and even though increasing longevity goes hand in hand with increasing vitality, not all additional years are actually healthy years. Moreover, the increasing demand for informal care must be met by a smaller number of relatives. Previous generations often included large families in which several siblings lived in the area where they grew up and could share the care for the older generations. Currently, many citizens in their fifties find little opportunities to share informal care tasks with siblings, either because they come from small families or because their siblings live far away. This lack of sharing opportunities may add to WFC [16,17,18].

Several studies have shown that WFC may negatively impact people’s quality of life [19,20], may give people an agitated feeling, cause stress, discomfort and sleep problems [21], which—especially if it continues over a longer period of time—may result in different medical problems, both physical and mental, and may even cause depression and burn out [17,22,23,24,25]. To reduce WFC and its undesirable consequences, a wide variety of institutional arrangements have been developed in order to help individuals and families reconcile the competing claims on their limited time. Usually, welfare state theories groups these arrangements into three categories [26,27,28]: arrangements that provide time (for instance, in the form of leave arrangements), arrangements that provide services (such as child care in day-care centres), and arrangements that provide families with financial means (such as child-related family allowances). Depending on the type of welfare state (liberal, social-democratic or corporatist/continental) [29,30] the focus can be more on public arrangements that enlarge citizens’ options to make their own choices regarding the arrangements they would like to make by providing them more money, on providing (child care) facilities that allow citizens to be active in the labour market while their children are cared for in a professional setting or on (leave) facilities that allow citizens to take up the care tasks themselves for some time [31,32]. Moreover, all these facilities can be provided and paid for either at the public level of the national government or the municipality where one lives or at the private level of the company where one is employed. In practice, European countries show a mix of facilities, although the focus between countries indeed differs [33,34]. In addition to these arrangements particularly designed to reconcile WFC, there are also all kinds of institutional arrangements that—although they are not primarily developed for this purpose—may also influence individuals’ opportunities to solve the WFC [35]. One could point to the tax system or to the educational system [36,37] where, in some countries, schools provide meals for the pupils, while in other countries, parents must pick up their children for lunch at home. Most salient, however, seem to be arrangements related to labour market flexibility, such as part-time work [38] and flexible working schedules [39]. Many of these arrangements are shaped at the company level [28]. As a result, working individuals and families have access to different arrangements to help them combine their labour market activities and care tasks, depending on their employer and in which country or municipality they live [28,39,40,41].

The limited set of international comparative studies on (the impact of) public facilities shows mixed results. For example, Stier [42] found that child-care availability reduced perceived WFC among women with children, but not among men. In addition, maternity leave policies reduced WFC among all women, but no effect was found for men. A study on nine EU-countries [43] concluded that temporary workers experience more work-life balance than workers with a permanent contract. Other studies have focused on the availability and/or use of specific arrangements, without studying the impact on WFC (for an overview, see [44]). Flexible working time arrangement hours may improve parents’ opportunities to bring their children to school in the morning and pick them up again later in the day. However, if labour market flexibility implies a high degree of uncertainty on when the employer calls on a worker to sign in, this complicates the combination of work and family life [45]. And the same may hold for the opportunity to work from home. On the one hand, it may help a worker to schedule his/her tasks in such way that it solves the intertemporal problem, on the other hand, it may blur the boundaries between work and private life and saddle up individuals with additional pressure from the idea that they have not done enough [46]. In the domain of psychology, there is a broad range of studies on the relationship between social support and work-family conflict [47,48,49]. In a seminal meta-study, French et al. [50] compared work-family support relationships with work-family conflict across different forms of support (behaviour, perceptions), sources (e.g., supervisor, co-worker, spouse) and types (instrumental, emotional). They included national contexts (cultural values, economic factors) as moderators, but no measures for actual government/public contributions to resolving work-family conflict.

Given the results from these different strands of research, in this study, we included both national facilities and provisions available at the organizational level as well as two indicators for the economic and emancipation climate in a country and our twofold goal was to analyse (1) the perceived level of work-life conflict in member states of the European Union and (2) the impact of national and organisational facilities that (aim to) support the combination of work and family life on perceived work-life conflict. More specifically, the impact of three national facilities was analysed: the availability of formal childcare; the amount of leave; the availability of long-term care. Regarding facilities at the organisational level, the main focus was on working time flexibility. We used data from the 2015 European Working Conditions Survey (EWCS), a large-scale survey among the working population in 35 European countries offering information on a whole series of items related to work, working conditions and arrangements at work, including perceived WFC. We combined this dataset with harmonised macro-data on the three work-family facilities. Due to reasons of availability of these macro-data, we limited our analyses to the 28 member states of the European Union. Thus, for each European citizen in our dataset, we know (a) how much WFC he or she experienced, (b) a series of personal characteristics, (c) a series of work characteristics and conditions and d) three national characteristics and facilities available to reduce WFC. As the data are multilevel, with working individuals (level 1) nested in countries with specific facilities (level 2), we conducted multi-level analyses. We conducted these analyses separately for women and men to evaluate possible differences. Our findings show that despite the national institutional differences, the intensity of WFC does not vary widely between European countries. In most countries, men experience less WFC than women, although the difference is small. Organizational arrangements and conditions are much more relevant for WFC than facilities offered by the government. In the next section, we go into more detail on the data and the analyses.

## 2. Materials and Methods

### 2.1. Sample

We used the European Working Conditions Survey (EWCS) 2015 of Eurofound [51], a five yearly cross-sectional survey with the aim to picture “everyday reality of men and women at work” [52]. EWCS focuses on a range of work-related topics, including employment status, work organisation, learning and training, working time duration and organisation, physical and psychosocial risk factors, health and safety, work-life balance and worker participation. Interviews were conducted face to face with the respondents being “representative of those aged 15 and over (16 and over in Bulgaria, Norway, Spain and the UK) living in private households and in employment who did at least one hour of work for pay or profit during the week preceding the interview” [53] (53:150). For each country, the sample consists of at least 1000 respondents. We focused on work-life conflict of the working population and therefore, excluded (a small group of) persons who, at the time of the survey, were temporarily inactive.

### 2.2. Dependent Variable

The dependent variable, work-family conflict, refers to perceived (in)compatibility of work and non-work (family care) activities [54]. EWCS contains five questions on WFC. A factor analysis shows that the five questions form one factor with high loadings of all components (see Table A1 in the Appendix A for questions and analysis). A factor analysis at the country level confirmed the single factor structure (reliability—indicated by Cronbach’s α-ranged from 0.7 in Croatia to 0.83 in Hungary and Romania). Per respondent, we computed a score on the WFC scale by adding up the scores on the five questions and dividing them by 5. To make the interpretation easier, we reversed the original scores so that high values indicate a higher incompatibility between work and family.

### 2.3. Independent Variables: Macro-Level

As indicators for work-family facilities, we included macro-level as well as micro-level variables. Macro-level variables were included to test welfare state theory which argues that public provisions by the government have the broadest impact on the reduction of work-life conflict as these facilities are usually available to all citizens [26,27,28,42]. The macro-level variables were formal childcare services, the amount of leave and the amount of formal long-term care. Formal childcare services are generally considered important facilities to support working parents. The indicator used was the share of children below the age of 3 that are taken care of in a formal arrangement (pre-school, childcare at centre-based services or childcare at a day-care centre). This is a commonly used indicator of child-care services for pre-schoolers [28,42]. Data on these shares per country are available in the database of Eurostat. Regarding leave, two Directives of the European Council oblige member states to offer working parents a minimum period of maternity and parental leave [55]. However, most countries also have national regulations in this field, which differ widely on aspects such as duration and payment. To compare countries, we followed [28] and calculated the amount of effective leave in weeks, which takes into account the length of the leave and the level of payment. Finding a proper indicator of formal support for informal care is more challenging, as harmonized data are not available [56]. We used the % of GDP spent on long-term care [57], assuming that higher expenditures imply more formal care, which could lower the burden and ease the WFC of working men and women that provide informal care. Data are available in the database of Eurostat. Unfortunately, this third variable is far less accurate than the ‘official’ and widely used variables for childcare and leave. While the latter two directly relate to potential solutions for the work-family conflict, the first is a more indirect measure. Following French et al. [50], we included two variables to account for the national economic and the national cultural context, respectively: GDP and female activity rate. GDP per capita measures a country’s financial wealth and might be considered a proxy for the opportunities to reduce work-family conflict. Female activity rate measures the share of women participating in the labour market, indicating to what extent society is used to (the problems following from) the combination of work and care tasks. The welfare state literature shows that countries which are more committed to gender equality and emancipation usually also show higher activity rates of women [58,59]. With already three variables included in the analyses measuring public facilities to mitigate work-family conflict directly, we expected the effect of what French et al. use as ‘moderators’ to be limited.

Table 1 provides an overview of the macro-level data per country and shows the variation over EU28. It should be noted that child care coverage and the % of GDP spent on long-term care are fairly strongly correlated (*r* = 0.73). The other variables are only moderately correlated. A Table A2 with correlations is included in the Appendix A.

### 2.4. Independent Variables: Individual Level

With regards to the micro-level, we included a series of variables. Flexible working arrangements are also considered relevant work-family facilities as they could solve the intertemporal problem of the combination of work and family care [26,27,28,35,38,39]. These arrangements are usually settled at the level of the firm [28]. EWCS includes several questions on such arrangements and provides individual respondents’ reports on the facilities available at the organisation where they are employed. We also included the number of working hours, the possibility to work flexible hours and the possibility to take some hours off for personal or family matters. The number of working hours is included to measure the pressure from the domain of work and is based on the question: ‘how many hours do you usually work per week in your main job?’. We categorized the answers into four groups and included the four categories in the analyses as dummies. These are short part-time (1–20 h per week; 1 = yes), long part-time (21–30 h per week; 1 = yes), fulltime (working 31–44 h per week; reference) and long fulltime (45 h or more per week). The possibility to work flexible working hours was included as a dummy (1 = yes). It is based on the question ‘how are your working time arrangements set?’. Working hours are considered flexible when a respondent can choose between several fixed schedules determined by the company, can adapt working hours within certain limits, or can determine working hours entirely on his/her own (1 = yes). In addition, we included a ‘time’ variable indicating a family-friendly working regime, based on the question: how easy it is for the respondent to take an hour or two off during working hours to take care of personal or family matters (dummy variable, 1 referring to the answer very or fairly easy).

We also included other work-related variables. One refers to having to work ‘asocial’ working hours, that is, at night and/or on Saturday and/or Sunday (1 = yes). We expected that working on such hours contributes to a higher level of WFC [63]. Occupation was included, as previous studies have shown that responsibilities belonging to working in high white-collar jobs increases WFC [42]. We created dummies, based on the International Standard Classification of Occupations categories; the group high white collar includes managers and professionals (1 = yes), low white collar includes clerical workers and service and sales workers (1 = yes) and blue collar occupations include technicians, skilled agricultural, forestry and fishery workers, plant and machine operators, and assemblers (= reference). We also took into account employment status, that is, 1 indicating if the respondent is self-employed and 0 if employed with an employer. Studies [64] have shown that self-employment might give flexibility but is also demanding and often results in higher levels of WFC. Finally, we included sector of work as a standard control (1 = private sector).

We expected that care responsibilities would result in more WFC. We used two indicators: presence of children and provision of informal care. We assumed that the presence of a child would increase WFC and we expected the effect to be stronger the younger the child is. Therefore, the presence of children in the household was categorized into three dummies: one indicating the presence of at least one child aged 6 or younger (1 = yes), one group indicating the presence of a child aged 7–18 (1 = yes) and one group without children below 18 in the household (reference group). EWCS includes questions on the involvement in activities other than work, including ‘caring for elderly/disabled relatives’. We used this as an indicator of providing informal care; possible answers were recorded in a dummy: 1 = yes and includes respondents who indicate to care for elderly/disabled relatives on a structural basis (either daily, several times a week or several times a month), 0 = no and includes respondents who provide this care ‘less often’ or ‘never’ or indicate ‘not applicable’. We also included age of the respondent, measured in years, and age squared, having a spouse or not (3 dummies: working spouse (1 = yes), non-working spouse (1 = yes) and no spouse (= reference)). Level of education was included as three dummies: low education (up until primary education; 1 = yes), medium education (secondary education; reference) and high education (tertiary and university education; 1 = yes). Table A3 in the Appendix A provides a table with descriptive information on the individual-level data.

### 2.5. Research Model 

We estimated WFC by taking into account variation within countries. Given the multilevel structure of the data with individuals nested within countries, we applied multilevel linear mixed models regression analyses (Source: The Statistical Package for the Social Sciences, version 24 [65]). Following similar studies in the field of welfare states [42,57,66], we estimated several random intercept models to investigate the impact of the macro-level variables while controlling for the micro-level variables. The basic model is
WFC_ij_ = *β*_0*j*_ + *β*_1_(MC_j_) + *β*_2_(X_ij_) + *β*_3_ (MC_j_ * X_ij_) + μ_0j_ + ε_ij_(1) where WFC_ij_ is the level of work-family conflict for individual i in country j, MC_j_ reflects the macro variables and X represents a vector of the (individual) demographic and work variables. In addition, cross-level interactions were added (see below for details). The *β* coefficients are fixed effects; for the individual-level variables, they indicate the effect on WFC of individual characteristics across countries. The effects of the macro-level variables can be interpreted as effects on WFC in an institutional context (countries) of individuals who are equal in terms of the individual-level variables included in the model. The random parameters are μ_0j_ (country level) and ε_ij_ (individual level) and indicate the difference from the means at the country resp. the individual level. We calculated the variance partition coefficient (VPC), which gives the proportion of the total residual variation that is due to the variation between countries. Firstly, a model with only the individual-level variables was examined with country as a group variable. In the next steps, we added the three variables for work-family facilities. We did separate analyses for each facility and a cross-level interaction between the facility and the group for which this facility is most relevant: the availability of child care and effective leave were combined with the presence of a child aged 6 or younger, whereas the level of formal long term care was combined with being an informal carer, Finally, one analysis included all three facilities and interactions. Female activity rate and GDP are included in all analyses. Following comparable multilevel studies [42,57,66], we report significance levels at 0.01, 0,05 and 0.10.

## 3. Results

Figure 1 provides an overview of the average perceived level of WFC by gender in the European member states. On a scale from 1 (lowest) to 5 (highest), the overall level of WFC is moderate. There is some variation between the countries, although rather limited. The highest level of WFC was found in the Southern-European countries Greece, Cyprus and Malta. The lowest level was found in Germany and Hungary. In most countries, women perceive higher WFC than men; although statistically relevant, the difference is small. In five countries, WFC is slightly higher for men: Czech Republic, Lithuania, Austria, Estonia, Croatia. For Austria and Croatia, the difference is not significant.

The multilevel analyses, which are presented in Table 2 and Table 3, confirm that differences in WFC between countries only account for a minor part of the total variation in WFC experienced by individuals in Europe. The largest part of this variation arises within countries. As expected, care obligations indicated by the presence of children within the household add significantly to WFC, both for women and men, and the effect is stronger for having children in the youngest age group. In addition, the provision of informal care adds significantly to WFC for both women and men.

With respect to the national facilities, we found that neither of these facilities have a major effect on individuals’ experience concerning WFC. Among women with children, we found a small effect of the length of the leave period: the more weeks of leave, the less WFC. A similar small effect in the opposite direction was found among women for the interaction term of the share of children in day-care centres in a country and the presence of children in the household. The latter effect disappeared when all three national arrangements were included jointly in the analysis. For women, we did not find any effect of the national expenditures on long-term care. Among men, we only found one effect of child-care: the higher the share of children in day care centres, the higher the experienced WFC among men with children present in the household. The somewhat surprising effects of child-care facilities on the level of WFC—although small—may result from the way we treated country differences in our analyses. Since we included only very few country variables in the analyses, this implies that all country variations were included in and clenched into these variables. One could imagine that modernity and lack of traditionalism in a country is partly reflected in the presence of day-care facilities. A less traditional division between male breadwinners and female homemakers is—as we stated in the introduction—also the major driver of the emergence of WFC. Thus, this could be one of the reasons behind the somewhat surprising relationship we found.

Working conditions and characteristics relating to job flexibility in terms of working hours contribute substantially to the explanation of the level of experienced WFC. People working part-time find it easier to combine work and care tasks than full-time workers, while among full-time workers, those working long hours experience more WFC than ‘regular’ full-time workers. Among part-time workers, we found similar differences: part-timers with a small part-time job experience even less WFC than workers holding a large part-time job. As mentioned in the introduction, the effect of flexible working hours is strongly debated in the literature. Our analyses show that flexible working hours do indeed contribute to WFC. We found this effect for both women and men. WFC can substantially be reduced by granting workers the opportunity to take a few hours off for personal or family reasons. This helps workers to solve urgent and occasional problems that arise in everyday life and can hardly be foreseen and directed in advance. As expected, working at inconvenient, irregular hours (at night, during the weekend) strongly adds to experienced WFC, again both for women and men.

Looking at personal characteristics, the table shows that managers and professionals (‘high white collar’) reported more WFC than lower white-collar and blue-collar workers. We found these effects for both women and men. Self-employed workers, both women and men, experience much more WFC than employees, which does not come as a surprise if one thinks of all the different dimensions of work self-employed workers are responsible for (the quality of the work one delivers, marketing, financial administration, etc.). Apparently, the (relative) freedom to choose your own hours, not only per day, but also over the year (which may, for instance, be helpful in synchronizing one’s work with school children’s holidays) does not offset the many demands on the time budget following from the multiple tasks.

Remarkably, women with a non-working spouse experience more WFC than women without a spouse or a working spouse. Although it would be logical to assume that the presence of a househusband would put off at least some of the time pressure for women, there are other forces at work that add to WFC. Although the EWCS does not provide any more details on this topic, one might consider that the possible role conflict following from the wife not completely fulfilling her ‘duties’ as a mother and the husband ‘forsaking’ his role as a breadwinner adds to WFC. Among men, those with a non-working spouse experience slightly more WFC than those with a working spouse. Apart from gender, a family with one non-working spouse has, on the one hand, more time available for care and family tasks, but on the other hand, also has to make end meets with less earnings than a two-earner family. Whether a potential lack of earnings outweighs a potential lack of time in its effect on experienced WFC will always depend on the particular circumstances of the family.

With respect to the background variables GDP and female activity rate as global indicators of the countries’ wealth and the extent to which a country is accustomed to two-earner families, the analyses show no significant effects of GDP and a small mitigating effect of higher female activity rates on work-family conflict among women. As mentioned previously, GDP may be too crude a measure to indicate a country’s opportunities for providing public facilities to reduce work-family conflict. Moreover, differing from French et al. [50], we also included direct measures for public facilities, which may cast most of the (small) effect of public facilities. The effect we found for female activity rate may indicate that a higher labour market participation rate among women and more dual earner families may result in other changes—not explicitly observed here—that make the reconciliation of work and family life somewhat easier for women. Another possibility may be that seeing other women struggling too may partly reconcile women with their own work-family conflict. Here, we should not forget that from an employer’s perspective, as well as national choices regarding childcare, leave and expenditures on long-term care set the stage on which they deploy their own organisational policies.

## 4. Discussion

In this article, we analysed (1) the perceived level of work-life conflict in member states of the European Union and (2) the impact of national and organisational facilities that (aim to) support the combination of work and family life on perceived work-life conflict. Our purpose was not to find the best explanatory model for work-life conflict.

We contributed to the literature by (eclectically) combining elements from different strands of literature, such as the sociologically and economically oriented welfare state literature, more individually oriented psychological studies and organizationally focused HR-studies. Secondly, in our empirical analyses, we combined macro-level facilities, work conditions and facilities determined at the organizational level as well as personal/family characteristics. Moreover, we added some background indicators at the national level that set the stage for organizations, individuals and families to make their choices dealing with work-family conflict. Thirdly, we not only payed attention to work-family conflict following from care responsibilities for (young) children, but also raised the issue of providing informal care. Finally, given the availability of the large dataset of the sixth European Working Conditions Survey, we were able to compare the 28 countries of the European Union, covering different types of welfare states and not having to limit ourselves to one specific country or one specific profession.

When evaluating the findings of our study, one should keep in mind that the results might be biased due to different forms of selectivity [32]. Logically, we could only include individuals in our study for whom we could calculate a value for work-life conflict. Thus, we could not incorporate, in our study, individuals who, at some point in time during their career, dropped out of the labour market because high levels of WFC resulted in burn out or made them decide to give up paid work in exchange for a more relaxed life. Neither could we include individuals who—as substantially large groups of women in different EU member still states do—dropped out of the labour market at the birth of their first child to be a full-time parent in order to prevent WFC. In addition, theoretically, one even could imagine some people refraining from being a parent because they did not feel up to facing the problem of combining work and family responsibilities. Although a majority of European citizens opts to be both in the labour market and have children, we do not know which (potential) role WFC plays in their decision not to combine both roles (any longer). For our results, this implies that the calculation of WFC is likely to be an underestimation of the ‘true’ degree of WFC in Europe.

For reasons of selectivity, we could unfortunately not include any income measure. Although welfare state theory points to the relevance of income as a means to arrange private solutions to solve work-family conflict, we did not include income in the analyses. The data from EWCS on income suffer from high non-response, which might be selective. Moreover, the information which is available refers to net income of the individual, not the household. Finally, as tax systems vary widely in Europe, net income is difficult to compare.

Since most variables included in the analyses have different dimensions (some are measured in time, others in percentages or simply ‘yes’ or ‘no’) it is not possible to compare the effects of, for instance, variables at the country level and indicators of flexibility of working hours. Thus, we cannot conclude whether arrangements at the national level are more or less effective in reducing WFC than arrangements at the organizational level. For the same reason, we cannot mutually compare national or organizational facilities. For variables at the national level, one could imagine trying to include all facilities in terms of euros. Apart from measuring problems, the gains in terms of comparability are undone by the losses in terms of accuracy, as the productivity and effectiveness of one euro spend on for instance childcare or leave differs widely between EU-countries [67,68]. As we mentioned previously, direct measures in terms of how many children are in childcare and how many people actually take up leave are more directly connected with the problem of solving work-family conflict than information on the budgets involved. However, there are two things we can say on the effectiveness. The first is that some countries together with the organizations within these countries provide packages of work-family facilities to their citizens that are (slightly) more effective in combatting WFC than the packages provided in other countries. Different combinations of national and organizational facilities, however, can result in rather similar scores regarding WFC. The second point is that arrangements at the national level usually apply to a broader range of (potential) users than organizational facilities which are normally restricted to the (families of the) organization’s employees. This implies that changes in country level facilities often have more impact than changes in arrangements supplied at the organizational level. Moreover, the provision of facilities at the organizational level may increase workers’ dependency on a particular employer. From an employer’s perspective, this may be an instrument to attract and retain workers. For employees, facilities at the organizational level may act as a ‘golden strings’, preventing them from making a transition to another employer. For the labour market as a whole, work-family facilities at the organizational level may reduce job mobility.

In addition to the limitations following from selectivity and lack of comparability between different (categories of) variables, we should mention three other limitations, which simultaneously constitute challenges for future research. The first follows from the self-reported nature of the items we used for the construction of our dependent variable, work-family conflict. People with the same or similar problems reconciling work and family life may report different levels of work-family conflict because of different (national) traditions, experiences and ambitions. That is why it may be useful from a policy perspective to collect more information on, for instance, individual and family time budgets. On the other hand, from the perspective of health consequences, it is the experienced pressure from work-family conflict that makes people sick and unhappy.

Another limitation to this study is that we could not include supportive conditions from the side of the family, such as the contribution of grandparents to childcare or household chores. The mere presence of a partner—which we included in our analyses—does not reveal much about the governance structure within a household [5]. A partner can be helpful and supportive, but can also add to work-family conflict through additional time claims (“I want to have dinner at seven” or “Why do we never go to the movies anymore?”). Therefore, future research might also benefit from taking stock of conditions and facilities at the household level that either contribute to higher or lower work-family conflict.

As a final limitation, we would like to point to the variables we included to picture the national background against which individuals and families have to solve their work-family conflict. Even though the variable female activity rate in particular actually contributed to the explanation of work-family conflict, once more, one could argue that it is only a distinct proxy for a whole range of different factors that adds to or diminishes work-family conflict. In some countries, parents struggle with school hours. In other countries, it is not school hours that cause problems, but the opening hours of shops or the hours at which one can go and see a doctor or the dentist. Therefore, it does not come as a surprise that working part-time and/or having the opportunity to take some hours off contributes to a reduction of work-family conflict. To see what is really relevant from the perspective of policy measures to reduce work-family conflict, it might be worthwhile to add variables for each country on school hours, opening hours of shops, medical services etcetera. This should give a more detailed perspective on the national context than a general variable such as female participation rate, which, by itself is—similarly to GDP—already an outcome measure.

Regarding future research on work-family conflict, it is clear that demographic developments in the field of increasing longevity and its consequences for the need and demand for care for the elderly will require much more attention than we gave the subject in our analyses. Hopefully, touching upon this issue here challenges other researchers to take up the gauntlet.

## 5. Conclusions

Countries of the European Union show only minor differences with respect to WFC experienced by their working citizens. In most countries, women show more WFC than men, although the difference is small. Caring for children and providing informal care clearly add to WFC, both for men and women. The relatively small country differences in WFC show that different combinations of national facilities and organizational arrangements together can have the same impact on individuals; apparently, there are several ways to realise the same goal of WFC reduction. Overall, we see higher scores on work-family conflict in countries with low female activity rates and limited childcare facilities, whereas they are lower in countries with high female activity rates and generous leave arrangements. However, the differences within countries are substantial depending on the different conditions and arrangements at the organizational level. Working less hours is beneficial from a perspective of WFC, despite the fact that working less hours also implies less earnings, which may also cause problems in the household, though of a different nature than that of the competition between work and care tasks. A small facility such as the opportunity to take a few hours off for personal or family reasons substantially decreases WFC. Irregular and flexible working hours and being self-employed increase WFC. From a policy perspective, our results suggest that the trend towards more flexibility that is dominating European labour markets and has already been under critical scrutiny from the perspective of insecurity and lack of incentives for mutual investments in knowledge and skills [69,70] can also be questioned from the perspective of its contribution to balancing work, care and family life. Finally, in view of the ageing European societies, it is important that future facilities not only consider the impact of care for children for WFC, but also include the impact of care for older relatives.

## Figures and Tables

**Figure 1 ijerph-16-04419-f001:**
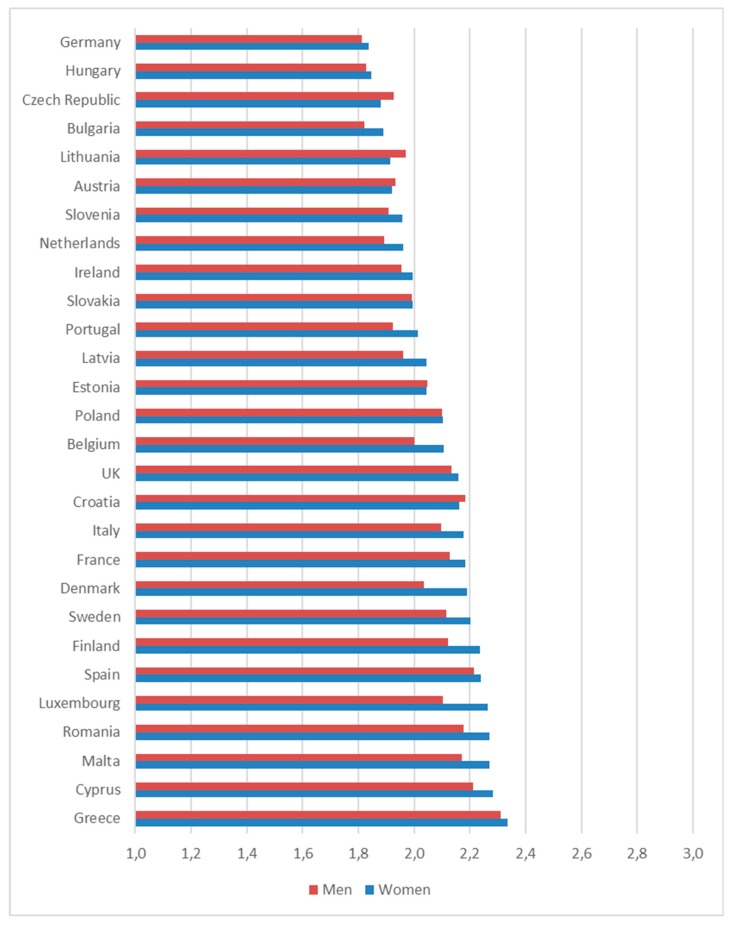
Perceived work-family conflict in EU28 by gender. (Source: [36])

**Table 1 ijerph-16-04419-t001:** Country values of the macro-level indicators.

Country	Child Care Coverage	Effective Leave in Weeks	Expenditure on Longterm Care	Female Activity Rate	GDP per Capita
Austria	22.3	68	1.5	70.9	37,500
Belgium	50.1	9	2.2	63.0	34,400
Bulgaria	8.9	52	0.0	65.4	13,700
Croatia	11.8	52	0.2	62.3	17,300
Cyprus	20.8	12	0.2	69.4	23,700
Czech Republic	2.9	22	0.9	66.5	25,300
Denmark	77.3	46	2.5	75.3	36,900
Estonia	21.4	76	0.4	73.0	22,000
Finland	32.5	38	1.9	74.4	31,700
France	41.7	10	1.7	67.3	30,700
Germany	25.9	60	1.8	73.1	36,000
Greece	11.4	9	0.2	59.9	20,200
Hungary	15.4	104	0.3	62.2	19,800
Ireland	30.6	20	1.6	65.2	51,900
Italy	27.3	14	0.9	54.1	27,700
Latvia	22.9	52	0.3	72.8	18,600
Lithuania	9.7	52	0.6	72.5	21,700
Luxembourg	51.8	8	1.4	65.6	77,300
Malta	17.9	10	1.4	55.5	27,200
Netherlands	46.4	10	2.7	74.7	37,800
Poland	5.3	46	0.4	61.4	19,900
Portugal	47.2	26	0.2	70.3	22,300
Romania	9.4	104	0.3	56.7	16,300
Slovakia	37.4	48	0.8	64.3	22,300
Slovenia	1.1	28	0.0	67.9	23,800
Spain	39.7	10	0.9	69.0	26,300
Sweden	64.0	57	2.9	79.9	36,400
United Kingdom	30.4	6	1.8	71.7	31,600

Sources: [55,60,61,62].

**Table 2 ijerph-16-04419-t002:** Results of two-level regression analyses (random intercept models) to explain the variable work-family conflict in EU28, women (standard errors in brackets).

Women’s Characteristics	Individual Level Variables Only		Facility: Child Care		Facility: Leave		Facility: Formal Long Term Care		All Facilities	
Level 1										
Age	0.023 ***	(0.003)	0.022 ***	(0.003)	0.022 ***	(0.003)	0.023 ***	(0.003)	0.022 ***	(0.003)
Age squared	0.000 ***	(0.000)	0.000 ***	(0.000)	0.000 ***	(0.000)	0.000 ***	(0.000)	0.000 ***	(0.000)
Child <6 at home (1 = yes)	0.259 ***	(0.016)	0.229 ***	(0.029)	0.307 ***	(0.024)	0.259 ***	(0.016)	0.294 ***	(0.039)
Child 7–18 at home (1 = yes)	0.144 ***	(0.014)	0.144 ***	(0.014)	0.144 ***	(0.014)	0.144 ***	(0.014)	0.144 ***	(0.014)
No child <19 at home (= ref)										
Provides informal care (1 = yes)	0.128 ***	(0.013)	0.129 ***	(0.013)	0.129 ***	(0.013)	0.123 ***	(0.022)	0.123 ***	(0.022)
Working spouse	0.010	(0.012)	0.010	(0.012)	0.010	(0.012)	0.010	(0.012)	0.010	(0.012)
Nonworking spouse	0.034 **	(0.017)	0.034 *	(0.017)	0.033 *	(0.017)	0.034 **	(0.017)	0.033 *	(0.017)
No spouse (= ref)										
High education	0.095 ***	(0.014)	0.095 ***	(0.014)	0.094 ***	(0.014)	0.095 ***	(0.014)	0.094 ***	(0.014)
Medium education (= ref)										
Low education	−0.014	(0.017)	−0.014	(0.017)	−0.014	(0.017)	−0.014	(0.017)	−0.014	(0.017)
High white collar	0.103 ***	(0.019)	0.103 ***	(0.019)	0.103 ***	(0.019)	0.102 ***	(0.019)	0.103 ***	(0.019)
Low white collar	0.005	(0.015)	0.005	(0.015)	0.004	(0.015)	0.004	(0.015)	0.004	(0.015)
Blue collar (= ref)										
Self employed (1 = yes)	0.137 ***	(0.019)	0.137 ***	(0.019)	0.136 ***	(0.019)	0.137 ***	(0.019)	0.137 ***	(0.019)
Private sector (1 = yes)	0.007	(0.012)	0.007	(0.012)	0.007	(0.012)	0.007	(0.012)	0.007	(0.012)
Short parttime	−0.251 ***	(0.015)	−0.251 ***	(0.015)	−0.251 ***	(0.015)	−0.251 ***	(0.015)	−0.251 ***	(0.015)
Long parttime	−0.120 ***	(0.016)	−0.121 ***	(0.016)	−0.121 ***	(0.016)	−0.121 ***	(0.016)	−0.121 ***	(0.016)
Fulltime (= ref)										
Long fulltime	0.250 ***	(0.017)	0.250 ***	(0.017)	0.250 ***	(0.017)	0.250 ***	(0.017)	0.250 ***	(0.017)
Flexible working hours (1 = yes)	0.065 ***	(0.013)	0.064 ***	(0.013)	0.065 ***	(0.013)	0.064 ***	(0.013)	0.064 ***	(0.013)
Possible to take hours off for family matters (1 = yes)	−0.253 ***	(0.011)	−0.253 ***	(0.011)	−0.253 ***	(0.011)	−0.253 ***	(0.011)	−0.253 ***	(0.011)
Works at asocial hours (1 = yes)	0.193 ***	(0.011)	0.193 ***	(0.011)	0.193 ***	(0.011)	0.193 ***	(0.011)	0.193 ***	(0.011)
Level 2										
% children in day care			0.004 **	(0.002)					0.004 *	(0.002)
Weeks of paid leave					−0.001	(0.001)			−0.001	(0.001)
Expenditure on longterm care							0.051	(0.042)	−0.003	(0.046)
Female activity rate			−0.010 **	(0.004)	−0.004	(0.004)	−0.008	(0.005)	−0.008 *	(0.004)
GDP per capita			0.000	(0.000)	0.000	(0.000)	0.000	(0.000)	0.000	(0.000)
Crosslevel interaction										
Children <6 at home X % children in day care			0.001	(0.001)					0.000	(0.001)
Children <6 at home X weeks of paid leave					−0.002 ***	(0.001)			−0.001 **	(0.001)
Expenditure on long term care X provides informal care							0.005	(0.016)	0.006	(0.016)
Beta	1.503 ***	(0.072)	2.042 ***	(0.287)	1.807 ***	(0.281)	1.932 ***	(0.312)	2.037 ***	(0.295)
σ_μ_^2^	0.019 ***	(0.005)	0.015 ***	(0.005)	0.017 ***	(0.005)	0.018 ***	(0.005)	0.015 ***	(0.005)
σ_ε_^2^	0.440 ***	(0.005)	0.440 ***	(0.005)	0.440 ***	(0.005)	0.440 ***	(0.005)	0.440 ***	(0.005)
−2 Log-Likelihood	33,163.01		33,209.58		33,208.17		33,202.60		33,237.31	
Variance partition coefficient	6.20%		4.91%		5.55%		5.79%		4.94%	
No. of women	16,325		16,325		16,325		16,325		16,325	
No. of countries	28		28		28		28		28	

Ref = reference group; *** *p* < 0.01 ** *p* < 0.05 * *p* < 0.10.

**Table 3 ijerph-16-04419-t003:** Results of two-level regression analyses (random intercept models) to explain the variable work-family conflict in EU28, men (standard errors in brackets).

Men’s Characteristics	Individual Level Variables Only		Facility: Child Care		Facility: Leave		Facility: Formal Long Term Care		All Facilities	
Level 1										
Age	0.033 ***	(0.003)	0.033 ***	(0.003)	0.033 ***	(0.003)	0.033 ***	(0.003)	0.033 ***	(0.003)
Age squared	0.000 ***	(0.000)	0.000 ***	(0.000)	0.000 ***	(0.000)	0.000 ***	(0.000)	0.000 ***	(0.000)
Child <6 at home (1 = yes)	0.173 ***	(0.017)	0.108 ***	(0.030)	0.215 ***	(0.025)	0.172 ***	(0.017)	0.152 ***	(0.039)
Child 7–18 at home (1 = yes)	0.070 ***	(0.016)	0.071 ***	(0.016)	0.071 ***	(0.016)	0.070 ***	(0.016)	0.071 ***	(0.016)
No child <19 at home (= ref)										
Provides informal care (1 = yes)	0.168 ***	(0.016)	0.168 ***	(0.016)	0.168 ***	(0.016)	0.194 ***	(0.025)	0.191 ***	(0.025)
Working spouse	0.093 ***	(0.014)	0.092 ***	(0.014)	0.092 ***	(0.014)	0.093 ***	(0.014)	0.092 ***	(0.014)
Nonworking spouse	0.108 ***	(0.016)	0.109 ***	(0.016)	0.108 ***	(0.016)	0.109 ***	(0.016)	0.109 ***	(0.016)
No spouse (= ref)										
High education	0.037 **	(0.015)	0.037 **	(0.015)	0.037 **	(0.015)	0.037 **	(0.015)	0.037 **	(0.015)
Medium education (= ref)										
Low education	0.015	(0.016)	0.014	(0.016)	0.014	(0.016)	0.014	(0.016)	0.014	(0.016)
High white collar	0.075 ***	(0.017)	0.075 ***	(0.017)	0.076 ***	(0.017)	0.075 ***	(0.017)	0.075 ***	(0.017)
Low white collar	−0.033 **	(0.013)	−0.033 **	(0.013)	−0.033 **	(0.013)	−0.033 **	(0.013)	−0.033 **	(0.013)
Blue collar (= ref)										
Self employed (1 = yes)	0.120 ***	(0.017)	0.120 ***	(0.017)	0.120 ***	(0.017)	0.120 ***	(0.017)	0.120 ***	(0.017)
Private sector (1 = yes)	0.010	(0.013)	0.010	(0.013)	0.011	(0.013)	0.010	(0.013)	0.011	(0.013)
Short parttime	−0.121 ***	(0.020)	−0.120 ***	(0.020)	−0.121 ***	(0.020)	−0.121 ***	(0.020)	−0.121 ***	(0.020)
Long parttime	−0.049 *	(0.025)	−0.049 **	(0.025)	−0.049 **	(0.025)	−0.049 **	(0.025)	−0.050 **	(0.025)
Fulltime (= ref)										
Long fulltime	0.260 ***	(0.014)	0.261 ***	(0.014)	0.260 ***	(0.014)	0.261 ***	(0.014)	0.261 ***	(0.014)
Flexible working hours (1 = yes)	0.127 ***	(0.014)	0.127 ***	(0.014)	0.128 ***	(0.014)	0.128 ***	(0.014)	0.127 ***	(0.014)
Possible to take hours off for family matters (1 = yes)	−0.260 ***	(0.012)	−0.261 ***	(0.012)	−0.260 ***	(0.012)	−0.260 ***	(0.012)	−0.260 ***	(0.012)
Works at asocial hours (1 = yes)	0.202 ***	(0.012)	0.202 ***	(0.012)	0.202 ***	(0.012)	0.202 ***	(0.012)	0.202 ***	(0.012)
Level 2										
% children in day care			0.002	(0.002)					0.002	(0.002)
Weeks of paid leave					−0.001	(0.001)			−0.001	(0.001)
Expenditure on longterm care							0.027	(0.041)	−0.014	(0.046)
Female activity rate			−0.005	(0.004)	−0.001	(0.004)	−0.003	(0.004)	−0.004	(0.005)
GDP per capita			0.000	(0.000)	0.000	(0.000)	0.000	(0.000)	0.000	(0.000)
Crosslevel interaction										
Children <6 at home X % children in day care			0.002 ***	(0.001)					0.002 **	(0.001)
Children <6 at home X weeks of paid leave					−0.001 **	(0.001)			−0.001 *	(0.001)
Expenditure on long term care X provides informal care							−0.024	(0.018)	−0.022	(0.018)
Beta	1.179 ***	(0.068)	1.513 ***	(0.288)	1.355 ***	(0.270)	1.404 ***	(0.303)	1.485 ***	(0.298)
σ_μ_^2^	0.015 ***	(0.004)	0.015 ***	(0.005)	0.015 ***	(0.005)	0.017 ***	(0.005)	0.015 ***	(0.005)
σ_ε_^2^	0.444 ***	(0.005)	0.443 ***	(0.005)	0.443 ***	(0.005)	0.444 ***	(0.005)	0.443 ***	(0.005)
−2 Log-Likelihood	31,649.53		31,696.78		31,700.08		31,691.50		31,726.04	
Variance partition coefficient	4.94%		4.96%		5.04%		5.45%		5.03%	
No. of men	15,525		15,525		15,525		15,525		15,525	
No. of countries	28		28		28		28		28	

Ref = reference group; *** *p* < 0.01 ** *p* < 0.05 * *p* < 0.10.

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
