# Peer review of "Work-Family Conflict in the European Union: The Impact of Organizational and Public Facilities"

_ijerph, 2019, doi:10.3390/ijerph16224419_

Round 1
Reviewer 1 Report
This is an interesting study. However, I have the following concerns:
A comprehensive literature review is missing. Without this part, it is hard to justify how this study advances our understanding in the field.More information is needed to understand the demographical characteristics of the sample used in this study. The authors had better summarize how the variables used in this study are measured in a section “measures”. The authors had better provide more detailed descriptions regarding their multilevel modeling (for example, the authors may refer to this article to follow their approach: https://www.mdpi.com/1660-4601/16/4/665). The contribution to the existing literature may not be significant, as the findings of this study are straightforward. Given that the EWCS is a big dataset, the authors had better find some moderators that may alter the impact of organizational and public facilities on WFC. The authors may refer to https://psycnet.apa.org/record/2017-56125-001
Or https://www.mdpi.com/1660-4601/16/17/3094 or https://europepmc.org/abstract/med/31535873 .
The discussion seems to be superficial. The authors had better link their findings to existing theories. The authors had better discuss the limitations of this study in the section “Conclusion”.
Hope my comments are helpful.
Reviewer 2 Report
This study uses a large survey on European countries to observe the effects of various variables affecting work-life conflicts.
The results show little effect from macro variables associated with work-family facilities, which is quite surprising.
Further discussion on the results of this study should be provided.
I have the following questions and comments.
Please do not use abbreviations such as WFC in Abstract. What is WLC on page 2? A more detailed description on the variables included in the ‘European working condition survey’ is needed. For example, if the household income level is included in the survey, the variable can be included in the analysis as an explanatory variable. What class does 'high white collar' refer to? Please clarify. Further explanation on the variables is needed in Table 2. For example, does ‘Age2’ refer to a squared variable? In addition, the interaction term is marked with '*' and should be marked with better indicator. A more detailed explanation is needed for the effect of the macro variable not being observed. The expenditure levels of countries on these policies are quite different. But if this costly policy doesn't have any effect on conflict level changes, where does all the money goes? In addition, the 'effective leave in weeks' variable varies greatly from country to country. I am not sure whether the variable is a good proxy for the work-family facilities.
Reviewer 3 Report
This article contributes to our knowledge about an important EU issue - work-family conflict, suffered by both women and men. As long as mothers find it difficult to combine work and family, it will be impossible for women to get ahead at the workplace – e.g, they will work fewer hours, earn less pay, qualify less for promotion. As long as fathers find it difficult to combine work and family, it will be very difficult for fathers to participate more in family life, and if they do, they too can suffer economically. EU countries depend on the dual-earner/dual-carer model for over all economic productivity, higher fertility, and children’s overall well-being.
This article also addresses what can solve work-family conflict –social policies or company policies particularly concerning flexible work. While many EU countries have exemplary policies to reduce work-family conflict, such as extensive paid parental leave, many do not. In some nations outside EU (e.g., the US), employees are more likely to secure family-friendly benefits from corporations than government.
This article also relies upon an important and reliable data set, the 2015 European Working Conditions Survey, supplemented with measures of country policies from Eurostat. They aim to analyze (p. 1):
“the level of WFC in the member states of the European Union by gender and to analyze to what extent different arrangements at the organizational level as well the public level help to reduce this.”
I believe the paper topic is important, but I have several concerns that I would like to see addressed.
In your introduction, you should probably mention that individuals and families can pay for services (which of course is a class-based problem); work-family conflict is not just solved by government or companies.
Please add a general description of who gets surveyed for the EWCS. E.g., All those employed 1 hour or more a week? All those in permanent jobs or are temporary workers included?
Measuring state policies
They examine childcare, leave, and informal care. But these are measured quite differently.
The childcare variables measures use - the percentage of children who use existing policy.
The leave variable measures the generosity of the policy (length, compensation).
The informal care variable measures how much is spent on long-term care.
This bothers me. Why not use how much is spent for each, probably the only way to make them similar? What difference might it make that the policies are measured so differently?
“informal care” is the label, yet the measure is how much the government spends on long-term care. To me, informal care is performed by family, not government. Can this be referred to as long-term care or formal elder care?
The measure for parenting is whether or not there is a child under 18 in the household. Is it possible to use a younger age to reflect the need to engage in more active parenting? The results might have been different.
Can you say something about whether or not the levels of wfc are considered high or not? E.g., were means over the average for the measure in every country?
If you would like to say something comparing men vs women in terms of work-family conflict, please do statistical tests. “Small” – does that mean still significant?
In your tables it appears that you use a .10 level of significance. This is not conventional and I would not recommend it. In fact, with a large data set, probably only .01 should be used. You should discuss your choice in the methods section.
I do not understand the tables, and since I do quantitative work and use multiple regression, I might not be alone. Consequently much more explanation of how results are presented within them is needed. For example – I am most familiar with seeing a dependent variable at the top of a regression table, and yet it doesn’t appear. I am most familiar with using standardized beta coefficients so comparisons between variable effects can be discussed, but you use unstandardized. Typically the amount of variance explained in the dependent variable (here: WFC) is presented so we can decide if the analysis really explained anything, but I didn’t see this.
This article REALLY needs to be edited for English by a native speaker and should not be published until it is. Funniest example –they indicate that plumbers fix roofs!
Round 2
Reviewer 1 Report
The authors have addressed my comments and I believe the current version meets the quality standard of IJERPH.
Reviewer 2 Report
I am happy with the changes made in the revised version.